# A Step Toward Quantifying Independently Reproducible Machine Learning Research

**Edward Raff**
Booz Allen Hamilton
`raff_edward@bah.com`
University of Maryland, Baltimore County
`raff.edward@umbc.edu`

## Abstract

What makes a paper independently reproducible? Debates on reproducibility center around intuition or assumptions but lack empirical results. Our field focuses on releasing code, which is important, but is not sufficient for determining reproducibility. We take the first step toward a quantifiable answer by manually attempting to implement 255 papers published from 1984 until 2017, recording features of each paper, and performing statistical analysis of the results. For each paper, we did not look at the authors code, if released, in order to prevent bias toward discrepancies between code and paper.

## 1   Introduction

As the fields of Artificial Intelligence (AI) and Machine Learning (ML) have grown in recent years, so too have calls that we are currently in an AI/ML reproducibility crisis [1]. Conferences, such as NeurIPS, have added reproducibility as a factor in the reviewing cycle or implemented policies to encourage code sharing. Many are pursing work centered around code and data availability as one of the more direct methods of enhancing reproducibility. For example, Dror et al. [2] developed a proposal to standardize the description and release of datasets. Others have proposed taxonomies and ontologies over reproducibility based on the availability of algorithm description, code, and data [3, 4]. Others have focused on building frameworks for sharing code and automation of hyper parameter selection in order to enable easier reconstruction of results [5].

While the ability to replicate the results of papers through open sourced code and data is valuable and should be lauded, it has been argued that releasing code is insufficient [6]. The inability to reproduce results without code availability may suggest problems with the paper. This may be due to the following: insufficient explanation of the approach, failure to describe important minute details, or a discrepancy between the code and description. We will call the act of reproducing the results of a paper without use of code from the paper's authors, *independent reproducibility*. We argue that for a paper to be scientifically sound and complete, it should be independently reproducible.

The question we wish to answer in this work is *what makes a paper independently reproducible*? Many have argued fiercely for different aspects of writing and publishing as critical factors of reproducability. Quantifiable study of these efforts is needed to advance the conversation. Otherwise, we as a community will not have scientific understanding that our work is addressing aspects of reproducibility. Gundersen and Kjensmo [7] defined several paper-properties of interest in regard to reproducibility. However, they *defined* a paper as reproducible purely as a function of the features without knowing if the selected features (e.g., method is described, data is available) actually impact a paper's reproducibility.

As a first step toward answering this question, we performed a study of 255 papers that we have attempted to implement independently. We developed the first empirical quantification about indepen-

dent reproducibility by recording features from each paper and reproduction outcome. We will review the entire procedure and features obtained in section 2. In section 3 we will discuss which features were determined to be statistically significant, and we will discuss the implication of these results. We will discuss the deficiencies of our study in section 4, with subjective analysis in section 5, and then conclude in section 6.

## 2    Procedure and Features

For clarity, we will refer to ourselves, the author of this paper, as the *reproducers*, distinct from the authors of the papers we attempt to independently reproduce. To perform our analysis, we obtained features from 255 papers. Inclusion criteria included papers that proposed at least one new algorithm/method that is the subject of reproduction, and papers where the first implementation and reproduction attempts occurred between January 1st 2012 through December 31st 2017. We chose varied paper topics based on our historical interest. No papers were included from 2018 to present, as some papers take more time to reproduce than others, which could negatively skew results for papers from the past year. If the available source code for a paper under consideration was seen before having successfully *reproduced* the paper, we excluded the paper from this analysis because at that point we are not a fully independent party. In line with this, any paper was excluded if the paper's authors had any significant relationship with the reproducers (e.g., academic advisor, co-worker, close friends, etc.) because intimate knowledge of communication style, work preferences, or the ability to have more regular communication could bias results. A paper was considered to be reproduced if the code for results were written by the reproducers, allowing the use of reasonable and standard libraries (e.g., BLAS, PyTorch, etc.), and the code reproduced the majority of claims from the paper.

Specifically, we regarded a paper as reproducible if the majority (75%+) of the claims in the paper could be confirmed with code we independently wrote. If a claimed improvement was measured in orders-of-magnitude, being within the same order-of-magnitude was considered sufficient (e.g., a paper claims 700x faster, but reproducers observe 300x). This same order-of-magnitude criterion comes from an observation that such claims are highly dependent upon constant factor efficiency improvements that may be had/missing from both the prior methods, and the proposed method being replicated. Presence or absence of these improvements can cause, apparently, "dramatic" impacts without fundamentally changing the nature of the contribution we are attempting to reproduce. When compared to other algorithms, we consider a paper reproduced if the considerable majority (90%+) of the new algorithm's rankings correspond to those found in the paper (e.g., the claim is that the proposed method was most accurate on 95% of tasks compared to 4 other models, we want to see our reproduction be most accurate on at least $95\% \cdot 90\% = 81\%$ of the same tasks, compared to the same models). As a last resort, we considered getting within 10% of the numbers reported in the paper (or better), or in the case of non-quantitative results (e.g., GAN sample quality), we subjectively compare our results with the paper to make a decision. We include this flexibility in specification to allow for small differences that can occur. While not common, we did encounter more than one instance where our independent reproduction achieved better results than the original paper.

After this selection process, we are left with 255 papers, of which 162 (63.5%) were successfully replicated and 93 were not. We note that this is significantly better than the 26% reproducibility determined by [7], who defined reproducibility as a function of the features they believed would determine reproduction. Below we will describe each of the features used. We attempt to catalog both features that are believed relevant to a paper's reproduction and features that should not be relevant, which will help us quantify if these expectations hold. We will use statistical tests to determine which of these features have a significant relationship with reproduction. An anonymized version of the data can be found at `https://github.com/EdwardRaff/Quantifying-Independently-Reproducible-ML`.

### 2.1    Quantified Features

We have manually recorded 26 attributes from each paper, which took approximately 20 minutes per paper to complete[1]. A policy for each feature was developed to minimize as much subjectivity as possible. Below we will review each feature, and how they were recorded, in order from least to

most subjective. Each feature was obtained based on the body of the main paper only, excluding any appendices (unless specified otherwise).

Features to consider were selected based on two factors: 1) would one reasonably believe the feature should be correlated with the ability to reproduce a paper (positive or negative) and 2) was the feature reasonably available with little additional work? This was done to capture as much useful information as possible while also avoiding limiting our study to items where *a priori* one might believe that a feature's relevance (or lack thereof) to be "obvious."

**Unambiguous Features:** Some features are not ambiguous in nature. A few are simple and innate properties that require no explanation. This included the *Number of Authors*, the existence of an *appendix* (or supplementary material), the number of *pages* (including references, excluding any appendix), the *number of references*, the *year* the paper was published, the *year first attempted* to implement, the venue *type* (Book, Journal, Conference, Workshop, Tech-Report), as well as the specific publication *venue* (e.g., NeurIPS, ICML). Many papers follow a progression from Tech-Report to Workshop to Conference to Journal as the paper becomes more complete. For any paper that participated in parts of this progression, we use the version from the most "complete" venue under the assumption that it would be the most reproducible version of the paper allowing us to avoid issues with double-counting papers.

We also include whether or not the *Author Replied* to questions about their paper. If any author replied to any email, it was counted as a "Yes". If no author ever replied, we marked it as "No." In all cases, every paper author was sent an email before marking it as "No." If a current email could not be found, we marked that the authors were not contacted.

**Mild Subjectivity:** We spend more time expounding on the next set of features, which had minor degrees of subjectivity. We state below the developed procedure we used to make their quantification practical and reproducible.

- *Number of Tables*: The total number of tables in the paper, regardless of the content of those tables. While tables usually contain results, they often contain a wide variety of content, and we make no distinction between them due to their frequency and variety.
- *Number of Graphs/Plots*: The total number of plots/graphs contained in the paper which includes scatter plots, bar-charts, contour-plots, or any other kind of 2D-3D numeric data visualization.
- *Number of Equations*: Due to differing writing styles, we do not use equation number provided by the paper, nor do we count everything that might be typed between LaTeX "$$" brackets. We manually reviewed every line of every paper to arrive as a consistent counting process[2]. Inline mathematics were only counted if the the math involved 1) two or more variables interacting (e.g., $x \cdot y$) or 2) two or more "operations" (e.g, $P(x|y)$ or $O(x^2)$). If only one "operation" occurred (e.g, $P(x)$ or $x^2$), it was not considered. Inline equations were counted only once per line of text, regardless of how many equations occurred in a line of text. Whole-line equations were always counted, regardless of the simplicity of the equation. If multiple whole lines were used because of equation length (e.g., a "$+$" ), it was counted as one equation. If multiple whole lines were used due to showing a mathematical step or derivation, each step counted as an additional equation. Partial deference was given to equation numbers. If every line of an equation received its own number, they were counted accordingly. If a derivation over $n$ whole lines received only one equation number, the equation was counted $\lceil n/3 \rceil$ times.
- *Number of Proofs*: A proof was only counted if it was done in a formal manner, beginning with the statement of a corollary or theorem, and included at least an overview of how to achieve the proof. A proof was counted if it occurred in the appendix or supplementary material. Derivations of update rules or other equations did not count as a proof unless the paper stated them as a proof. This was done as a practical matter in reducing ambiguity and the process of collecting the information.
- *Exact Compute Specified*: If a paper indicated any of the specific compute resources used (e.g., CPU GHz speed or model number, GPU model, number of computers used), we considered it to have satisfied this requirement.
- *Hyper-parameters Specified*: If a paper specified the final hyper-parameter values selected for each dataset or the method of selecting hyper-parameters (e.g., cross validation factor) and the value range (e.g., $\lambda \in [1, 1000]$), we consider it to have satisfied this requirement. Simply stating that a grid-search (or similar procedure) was used was not sufficient. If a paper introduced multiple

hyper-parameters but only specified how a sub-set of the parameters where chosen, we marked it as "Partial".

- *Compute Needed*: We defined the compute level needed to reproduce a paper's results as needing either a Desktop (i.e., $\leq$ \$2000), a consumer GPU (e.g., an Nvidia Geforce type card), a Server (used 20 cores or more, or 64 GB of RAM or more), or a Cluster. If the compute resources needed were not explicitly stated, this was subjectively based on the computational complexity of the approach and amount of experiments believed necessary to reach reproduction. We stress that this compute level was selected based on today's common compute resources, not those available at the time of the paper's publication.
- *Data Available*: If any of the datasets used in the paper are publicly available, we note it as having satisfied this requirement.
- *Pseudo Code*: We allow for four different options for this feature: 1) no pseudo code is given in the paper, 2) "Step-Code" is given, where the paper outlines the algorithm/method as a sequence of steps, but the steps are terse and high-level or refer to other parts of the paper for details, 3) "Yes", the paper has some pseudo code which outlines the algorithm at a high level but with sufficient detail that it feels mostly complete, and 4) "Code-Like", the paper summarizes the approach in great detail that is reminiscent of reading code (or is in fact code ).

**Subjective:** We have a final set of features which we recognize are of a significantly subjective nature. For all of these features, we are aware there may be significant issues, and in practice, any alternative protocol would impose its own different set of issues. We have made the choices in an attempt to minimize as many issues as possible and make the survey possible. Below is the protocol we followed to reduce ambiguity and make our procedure as reproducible as possible for future studies, which will help the reader fully understand our interpetation of the results.

- *Number of Conceptualization Figures*: Many papers include graphics or content for which the purpose is not to convey a result, but to try to convey the idea / method proposed itself. These are usually included to make it easier to understand the algorithm, and so we identify them as a separate item to count.
- *Uses Exemplar Toy Problem*: As a binary "Yes"/"No" option, did the paper include an exemplar toy problem? These problems are not meaningful toward any application of the algorithm, but they are devised to show specific behaviors or create demonstrations that are easier to reproduce / help conceptualize the algorithm being presented. These are often 2D or 3D problems, or they are synthetically generated from some specified set of distributions.
- *Number of Other Figures*: This was a catch-all class for any figure that was not a Graph/Plot, Table, or Conceptualization Figure as defined above. For most papers, this included samples of the output produced by an algorithm or example input images for Computer Vision applications.
- *Rigor vs Empirical*: There have been a number of calls for more scientific rigor within the ML community[8], with many arguing that an overly empirical focus may in fact slow down progress [9]. We are not aware of any agreed upon taxonomy of what makes a paper "rigorous". Based on the interpretation that rigor equates to having grounded understanding of why and how our methods work, beyond simply showing that they do so empirically, we develop the following protocol: a paper is classified as "Theory" (read, rigorous), if it has formal proofs, provides mathematical reasoning or explanation to modeling decisions, or provides mathematical reasoning or explanation to why prior methods fail on some dataset. By default, we classify all other papers as "Empirical." However, if a "Theory" paper also includes discussion of practical implementation or deployment concerns, complete discussion of hyper-parameter setting such that there is no ambiguity, ablation studies of decisions made, or experiments on production datasets, we consider the paper "Balanced" as having both theory and empirical components.
- *Paper Readability*: We give each paper a readability score of "Low", "Ok", "Good", or "Excellent." To minimize subjectivity in these scores, we tie each to the amount of times we had to read the paper in order to reach a point where we felt we had the proposed algorithm implemented in its entirety, and the failure to replicate would be a matter of finding and removing bugs. The score of "Excellent" means that we needed to read the paper only once to produce an implementation, "Good" papers needed two or three readings, "Ok" papers needed four or five, with "Low" being six or more reads through the paper[3].
- *Algorithm Difficulty*: We categorize the difficulty of implementing an algorithm as either "Low", "Medium", or "High." We grounded this to lines of code for any paper successfully implemented

or which made its implementation available online. For ones never successfully implemented and without code, we estimated this based on our intuition and experience on where the implementation would have landed based on reading the paper. "Low" difficulties could be completed in 500 lines of code or less, "Medium" difficulty between 500 and 1,500 lines, and "High" was > 1,500 lines. In these numbers we assume using common libraries (e.g., auto-differentiation, BLAS, etc.).

- *Primary Topic*: For each paper we tried to specify a single primary topic of the paper. Many papers cover different aspects of multiple problems, making this a challenge. We adjusted topics into higher-level categories so that each topic had at least three members, so that we could do meaningful statistics. Topics can be found in the appendix.
- *Looks Intimidating*: The most subjective, does the paper "look intimidating" at first glance?

# 3 Results

Our features are either numeric or categorical. For each numeric feature (except the number of pages and number of authors), we normalized the value by the number of pages in the paper. Longer papers naturally have more space to include more equations, figures, etc., and this was done to make all papers more directly comparable. For numeric features we used the non-parametric Mann–Whitney U [10] test to determine significance. A Shapiro-Wilk test of normality [11] confirmed that none of our features would have been appropriate for use with a Student's t-test and so the non-parametric testing is preferred. For all categorical features, we used a Chi-Squared test [12] with continuity correction [13]. In our analysis we will also examine relationships between some of our categorical features and other numeric features for suspected relationships. We will continue to use non-parametric tests for robustness/conservative estimates of significance, relying on the Kruskal-Walls [14] for ANOVA testing and the Dunn test [15] for post-hoc analysis. JASP was used to compute all statistical tests [16]. In Table 1 we show the results for deciding which of our 26 features were correlated with a paper's reproducibility. Tables and graphs of all the features are too numerous to fit in the main paper, and will be found in the appendix.

We begin by noting that the *year* a paper was published or the year that we first tried to implement the paper were not correlated with successful reproduction. The concerns of a reproducibility crisis would generally imply that the issue is a recent one. However, the year a paper was published is not correlated

Table 1: Significance test of which paper properties impact reproducibility. Results significant at $\alpha \leq 0.05$ marked with "*".

| Feature | p-value |
| --- | --- |
| Year Published | 0.964 |
| Year First Attempted | 0.674 |
| Venue Type | 0.631 |
| Rigor vs Empirical* | $1.55 \times 10^{-9}$ |
| Has Appendix | 0.330 |
| Looks Intimidating | 0.829 |
| Readability* | $9.68 \times 10^{-25}$ |
| Algorithm Difficulty* | $2.94 \times 10^{-5}$ |
| Pseudo Code* | $2.31 \times 10^{-4}$ |
| Primary Topic* | $7.039 \times 10^{-4}$ |
| Exemplar Problem | 0.720 |
| Compute Specified | 0.257 |
| Hyperparameters Specified* | $8.45 \times 10^{-6}$ |
| Compute Needed* | $8.75 \times 10^{-5}$ |
| Authors Reply* | $6.01 \times 10^{-8}$ |
| Code Available | 0.213 |
| Pages | 0.364 |
| Publication Venue | 0.342 |
| Number of References | 0.740 |
| Number Equations* | 0.004 |
| Number Proofs | 0.130 |
| Number Tables* | 0.010 |
| Number Graphs/Plots | 0.139 |
| Number Other Figures | 0.217 |
| Conceptualization Figures | 0.365 |
| Number of Authors | 0.497 |

with successful reproduction, with the oldest paper being from 1984. This would suggest that independent reproducibility has not changed over time. Depending on one's perspective, we could argue that there is not a reproducibility crisis, or that one has been ongoing for several decades. It is important the reader qualify this statistical result with the fact that the year of paper publication in our study is not evenly distributed over time, with the majority of papers occurring in between 2000 through 2017.

To our study's benefit, the year first attempted for reproduction was not significant. If our success was correlated with time (as one might expect in advance— with skill increasing with experience), we would worry about this skewing our results. This appears to not be an issue, removing a potential problem from our results.

### 3.1 Significant Relationships

There were ten variables that are significantly correlated with a paper's reproducibility. Of them, Number of Tables, Equations, Compute Needed, Pseudo-Code, and Hyper-parameters Specified are the least subjective variables which were significant.

Readability had the strongest empirical relationship, which on its face is not surprising. Note that by our definition, Readability corresponds to how many reads through the paper were necessary to get to a mostly complete implementation. As expected, the fewer attempts to read through a paper, the more likely it was to be reproduced. For "Excellent" papers, we were always able to reproduce results. Exact counts can be found in Table 11. Based on these results we argue the importance of clear and effective communication of implementation details, which may often be neglected. This neglect may come from forced page limits or a preference towards other, competing factors (e.g., preferring figures/results that better show the method's value, at the cost of method details). We suspect that a factor in this is page limits which we test by proxy via paper page length. A Krusal-Wallis test confirms the significance of paper length in pages ($p = 0.035$). A Dunn post-hoc test shows that "Low" Readability papers are the statistically significant source of this relationship, which are 3.17–5.67 pages shorter than the other Readability types. As a field that has historically focused on open-access and online availability, and with the decreasing relevance of paper conference and journal distributions, our study suggests that raising page limits on papers, and adding technical algorithmic details as an explicit review factor, could aid in increasing the reproducibility of papers.

Two factors we expected to be related to a paper's Readability (as we have defined it) are the Algorithm's Difficulty and the presence of Pseudo-Code, both of which are significant factors and have a statistically significant relationship with Readability. If we look at Table 12, we see that Pseudo-Code has a complicated relationship with Reproduction.

Table 2: Relationship between use of Pseudo Code and Readability

| Pseudo Code | | Low | Ok | Good | Excellent |
|---|---|---|---|---|---|
| | | \multicolumn Paper Readability | | | |
| No | Actual | 22.00 | 10.00 | 23.00 | 24.00 |
| | Expected | 23.24 | 17.66 | 24.16 | 13.94 |
| Step-Code | Actual | 29.00 | 15.00 | 7.00 | 6.00 |
| | Expected | 16.76 | 12.74 | 17.44 | 10.06 |
| Yes | Actual | 21.00 | 28.00 | 39.00 | 14.00 |
| | Expected | 30.00 | 22.80 | 31.20 | 18.00 |
| Code-Like | Actual | 3.00 | 4.00 | 9.00 | 1.00 |
| | Expected | 5.00 | 3.80 | 5.20 | 3.00 |

Highly Detailed "Code-Like" descriptions are more reproducible, but having "No" pseudo-code is also positively related with reproduction. Based on these results papers which can effectively describe their algorithms without pseudo-code are communicating the information in another way, but papers with "Step-Code" do an inadequate job at this task. Examining the relationship between Pseudo-Code and Readability in Table 2 supports this, where we see that using Step-Code is biased toward lower readability. This also makes sense in abstract, as step-code often requires one to repeatedly reference different parts of a paper. The relationship between an Algorithm's difficulty is more direct and intuitive, Table 10 showing that reproducibility decreases with difficulty.

It is also interesting to note how Rigor vs Empirical is correlated with reproducibility. One may have expected papers that focus on proving their methods correct would be the most reproducible. In Table 8 we can see that papers that are "Empirical" or "Balanced" both have higher than expected reproduction rates, while "Theory" oriented papers have lower than expected. These results would seem to suggest that empiricism is intrinsically valuable for reproduction on the micro scale of individual papers. This does not contradict any of the concerns about long-term behaviors and results that are side effects of overly-empirical issues discussed by Sculley et al. [9], such as new methods being inappropriately considered due to ineffectively tuned baselines and lack of ablation studies. We take this result as a further indication that rigor cannot just be math or learning bounds for their own sake, but that the practical relevance and execution of any theorems must be at the forefront in all papers[4].

Unfortunately, the primary topic of a paper was found to be a significant factor for independent reproducibility. We were not able to reproduce any Bayesian or Fairness based papers. We had a higher-than expected success in implementing papers about Deep Learning and Search/Retrieval. We, the reproducers, are not experts in all of the primary topic areas listed, and so we advise against

extrapolation from this particular result. This leads to interesting questions regarding reproduction from inside/outside an expert peer group and when one qualifies as an expert in a general topic area. We hope to explore these questions further in future work.

Both Number of Tables and Hyper-parameters were positively correlated with reproducibility. The more tables included in a paper, or the more parameters specified, the more likely the paper was to be reproducible. This is not a surprising result for the Hyper-parameters case, and supports the emphasis the community has placed on this factor [5]. It is somewhat peculiar that Tables are significant, but Graphs/Plots are not as both convey primarily numeric information to the reader. We suspect that the ability for the reader to quickly understand the exact value/result from a table is the differentiating factor as it gives a target to meet and measure against. While a plot/graph may describe overall behavior, it may not readily avail itself to quickly extracting a hard number and using it as a goal.

The Number of Equations per page was *negatively* correlated with reproduction. Two theories as to why were developed based on our experience implementing the papers: 1) having a larger number of equations makes the paper more difficult to read, hence more difficult to reproduce or 2) papers with more equations correspond to more complex and difficult algorithms, naturally being more difficult to reproduce. A Kruskal-Wallis ANOVA reveals that the readability hypothesis is significant ($p = 0.001$) but not the difficulty hypothesis ($p = 0.239$). Following with a Dunn post-hoc test shows that papers which have "Excellent" readability have fewer equations per page (2.25 eq/pg) than the others, as the source of the significant ($p \leq 0.002$) relationship. There are no significant differences between papers of "Low," "Ok," and "Good" readability (3.91, 3.60, 3.78 equations per page respectively), leading us to postulate that the most readable and reproducible papers make careful and judicious use of equations.

Our last paper-intrinsic property is *Compute Needed*, which could be a "Desktop", "GPU", "Server", or "Cluster". In the time that these papers were implemented, we have had access to all four compute levels to varying degrees. Looking at Table 18, we see the use of a Cluster or GPU are the ones that depart from expectations. Despite having access to cluster resources, we have never successfully reproduced a paper that needed such resources. At the same time, we have a higher reproduction rate for works that require a GPU. Our suspicion is that frameworks such as PyTorch and Tensorflow, which make use of GPUs relatively easy, have been converging toward an effective paradigm for using that kind of resource. These libraries make it easier to reproduce current papers and historical ones that lacked such advanced tools, which then inflates reproduction rate. While frameworks like Spark exist for distributed computation, they may not be sufficiently developed for Machine Learning use cases to ease replication. Another alternative hypothesis for Cluster reproduction failure is that the details of how a cluster is organized, with interconnects, job scheduling, and more sophisticated code, are increasing the reproduction barrier and lack necessary details. We do not have sufficient information to confirm or reject these hypotheses, but we encourage others to consider them as avenues for study.

This leaves us with the last significant result, which is not a property of the paper itself: whether the paper authors reply to questions about their paper. We reached out to the authors of 50 different papers and had a reply rate of 52%. Table 7 which shows that replying was the most individually predictive attribute studied. In the 24 cases where the author did not respond to questions, we succeeded in replication only once. For the 26 cases where they did reply, we succeeded 22 times. While this result demonstrates the importance of corresponding with readers, it gives credence to the idea of a non-stationary and "living" paper where updates may be made over time to address questions and concerns. Such is possible today with arxiv.org and distill.pub, and provides quantifiable evidence that their ability to update articles is a meaningful and powerful tool toward reproducibility (if leveraged). Other confounding hypotheses exist as well, such as receiving a reply increasing the motivation of the reproducers, or the nature of a discussion that is not constrained to a paper's limitations may also impact reproduction rates.

## 3.2 Interesting Non-Significant/Negative Results

While we have already discussed some non-significant results as they relate directly to significant ones above, we also want to highlight interesting non-significant results. In particular, we expected *a priori* that the use of Conceptualization Figures and Exemplar Problems would be significant predictors, as we have found them useful in our personal experiences both to understand the algorithm, and as an initial test-bed to confirm an algorithm was working to a minimal degree. Yet neither are significant. We also find that neither have a relationship with a paper's Readability ($p \geq 0.476$).

These results give us pause regarding our assumptions about what makes a "good" reproducible paper, and reinforce the importance of quantifying these important questions.

A positive indicator is that Venue (e.g., NeurIPS vs PKDD) had no significant impact, nor did Venue type (e.g., Workshop vs Journal). This result would seem to imply that the same issues and successes are occurring across most academic levels, though selection bias may play a role in this result.

The non-significance of including an appendix is of note given our results that the papers which are hardest to reproduce ("Low" Readability) are shorter on average. There is no significant difference between a paper's readability and the presence of an appendix ($p = 0.650$), which implies that appendices are not sufficient means of circumventing page limits at conference/workshop venues.

We found it interesting that whether or not the papers' authors released their code has no significant relationship with the paper's independent reproducibility. Before analysis, we could see hypotheticals that would cause correlations in either direction. Authors who release code might include less details in the paper under the assumption that readers will find them in the code itself. Conversely, one might imagine that authors who release code care more about reproduction and would include more of the necessary details. With more conferences encouraging code availability as a reviewer criteria, we would not necessarily expect any change in independent reproducibility from this change in isolation (impacts on cultural changes induced being a question beyond our scope).

## 4    Study Deficiencies

While we have taken the first step toward studying and quantifying factors of reproducibility, we must also acknowledge deficiencies in our study. Most apparent are a number of potential biases. The papers under consideration have a selection bias based on interest and filtering from consideration any paper where we had previously looked at released source code. More importantly, all papers were attempted by just this paper's author. So while we have a large sample size of papers, we have a low sample size of implementers. It is entirely possible that those with a different background in education, training, career, and interests, would find different papers easy or difficult to reproduce.

Because we are the sole reproducer, all the results must be taken with consideration conditioned on our background, and the origin of this work. A majority of attempted reproductions where in pursuit of contribution to a machine learning library that we are the author of, JSAT [17]. As such, we focused initially on a number of more common and widely used algorithm. These methods had already been independently reproduced by others many times, and alternative materials (e.g., lecture notes) were available to provide guidance without consulting code written by others. Further papers where spurred by our personal interest in what we considered useful for such a library, and our own personal interests (historical interests including nearest neighbor algorithms, linear models, and kernel methods). Such well known works do not make the majority of reproduction attempts, but they make up a sizable sub-population of the methods we attempted for JSAT, and so may skew results.

Our study is also limited by our own historical records. The use of paper cataloging software to take notes and record information made this study possible, but it also limits our study to the recorded notes and what can be re-derived from the paper itself (e.g., number of pages).

Towards improving upon the number of implementers and recording information, we hope to encourage extensions to projects such as the ICLR Reproducibility Challenge. A communal effort to standardize on an initial set of paper features, keep track of time and resources spent on reproduction, and information about the reproducers (years of experience, education, and background) may allow for a richer and more thorough macro study of reproducibility in the future. A design constraint we would like to include in such a system is differential privacy so that it is not known which individual papers are having reproduction difficulties. We have intentionally avoided identifying papers to avoid any perceived "naming and shaming", as our or other attempts in isolation should not be seen as conclusive statements on any individual paper's lack of reproduction.

In our experience attempting to reproduce these papers, we also note a failure in the framing of the problem: that a paper is reproducible or not. Depending on the paper, differing levels of resources and even teams may be necessary for reproduction. As a point of reference, the longest effort toward reproduction we studied took 4.5 years of (non-continuous) effort to finally reproduce the results. In this light, it may be better to model reproduction as a kind of survival analysis conditioned on properties of the implementer(s). A paper "survives" as the implementers attempt reproduction and

"dies" once successfully reproduced (or "lives" forever if never reproduced). Viewed in this light, we may ask: what environmental factors (e.g., libraries like PyTorch, Scikit-Learn, compute resources) impact survival rates and times, and should the necessity of code release be a function of survival time? A real life example of this is playing out now, as people attempt to reproduce OpenAI's recent GPT-2 results[5], where information and data was intentionally withheld due to security concerns.

An important factor not included in our analysis are the authors of a paper, which has a direct impact on writing style, topic, and other factors. Subjectively we note that there are authors whose work we regularly fail to reproduce and ones we regularly succeed in reproducing, even when both make code available. Study of how the backgrounds and styles of both authors and implementers interact and impact reproduction seems to be a valuable line of inquiry, but it is beyond our current scope and requires additional thought and consideration.

We also note that the most significant factors in reproducibility are the most subjective factors. While we endeavored to reduce the impact of subjectivity as much as possible with our stated protocols, this indicates that more work is warranted in developing more objective measures that are related to these subjective factors, or using communal effort to reach a distributional determination on these subjective factors, for future studies.

## 5    A Subjective Recall of Non-Reproduction

We did not record the believed reason for failure to reproduce, although this would have been valuable information. We hope that this will be noted by others in the future, but for now we recount a subjective summary of the primary reasons we felt a paper could not be reproduced. We note that part of our belief in the below list stems from our efforts to email papers' authors when attempting to independently reproduce their works, in which we are often seeking information that would elucidate the below issues:

1. Unclear notation or language. A component of the algorithm is explained, but not in a way easily understood by the reproducers, or was ambiguously specified.
2. Missing algorithm step or details, a step was completely left out of description.
3. Many papers would specify loss functions or other equations for which the gradient needed to be taken, but not detail the resulting gradients. Depending on the functions and math involved re-deriving was non-trivial, and our results did not match.
4. Missing hyper-parameters, or similar nuance details. The reproducers believe we have an implementation accurate to what was described, but some "minor" detail was not specified and makes a big difference in results.

We avoided in this paper any attempt to imply or cast doubt on the veracity of any individual paper. In our experiences through this work, we have rarely had suspicion that the results of a paper were false or the result of serious flawed implementation, and thus could never be reproduced.

## 6    Conclusions

In this work we have conducted the first empirical study of what impacts a paper's reproducibility. We suspect this will lead to considerable debate about the meaning of results, and we hope to spur further quantifiable studies. Based on our results, we find that paper reproduction rates have not changed (in a statistically significant way) over the past 35 years. Papers of a more empirical nature tend to be more reproducible, as are ones that include factors relevant to implementation details — though simply including Pseudo-Code is not sufficient. Our study indicates papers with fewer equations and more tables tend to be more reproducible, and that there is a potential latent issue in reproduction when cluster computing becomes a requirement.

## Acknowledgements

I would like to thank Jared Sylvester, Arash Rahnama, Charles Nicholas, Cynthia Matuszek, Frank Ferraro, Ian Soboroff, and Ashley Klein, who all provided valuable discussion and feedback on this work through its formation to completion.

## Footnotes

[1]Not done in a continuous run. Feature collection, and paper selection, and total time preparing the study data took approximately 6 months.

[2]Not all papers have LaTeX available, and older papers are often scanned making automation difficult.

[3]This information was obtained from our own record keeping over time and paper-organizing software

[4]This would not apply for pure theory papers, and we remind the reader that all papers in this study proposed and evaluated some new algorithm, and thus do not fall into a pure theory category.

[5]`https://openai.com/blog/better-language-models/`

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
