[Supplementary Material · Appendix.pdf]

## A  Some Preemptive Responses to Questions

In preparation of this manuscript, I have sought advice and feedback from a number of colleagues. These discussions have resulted in a few common questions of related themes. As such, many do not necessarily belong in the main content of an academic paper. Here I will list and preemptively answer the common ones in hope of aiding the reader in better understanding this work, the context around it, and the potential biases that may exist in the results as a function of my personal background.

### A.1  Why Where You Recording Information About Papers?

Before I started working on JSAT, or had even learned about machine learning, I had a side project implementing an arbitrary precision math library[6]. I worked on this library for four years, and implemented a number of algorithms for computing different decomposition, mathematical constants, common functions (e.g., Fibonacci numbers), complex numbers, all at arbitrary precision. As part of this I began to read and implement a number of papers for these techniques. As time went on, and I occasionally found and discovered bugs in my previous implementations, I grew frustrated in the bug fixing process. Fixing bugs for these more involved methods required me to re-understand and find previous papers, which I was not good at. As a result, when I started JSAT, I began keeping notes to myself from the onset. I again intended it to be a multi-year and long term project, and wanted to avoid repetition of previous failures.

### A.2  Why Where You Implementing so Many Papers?

Early in my computer science career, I had forgone most all advanced math courses — taking the bare minimum to get my degree, with the exception of a numerical analysis course that I thought would be relevant to my arbitrary precision math library mentioned above. I had instead focused on taking many more CS courses until I happened to take a machine learning class and became enamoured with the concept and the field. This left me with a situation where I wanted to get into a domain that was math heavy, but my skills had long languished. While I personally felt I have always understood an algorithm or technique best once I can implement it, I came to rely on implementing an algorithm as a crutch to my lesser mathematical skills. When encountering some mathematical notation or concept I did not well understand, I could simply enumerate all the options I thought might be correct, and see which one eventually worked. This has continued far longer than I would like in many ways and so have continued to attempt to implement papers I want to understand as the fastest way for me to come to a functional understanding of a paper.

## B  Statistical Test assumptions

In Table 3, we perform a normality test, which confirms that all of our numeric features deviate significantly from a normal distribution, making a standard Student's t-test inappropriate for hypothesis testing.

The Mann-Whitney test assumes that the variance of the two distributions under test are equal. We can see from Table 4 that this again holds for all of our numeric features, with the exception of the Year of the publication and the total number of pages. If we instead preformed a Welch test, which does not have the equality of variance assumption, we still arrive at the conclusion that Year ($p = .554$) and number of Pages ($p = 0.134$) do not have any significant relationship with reproducibility. The pages variable is also impacted by a few outliers (the most extreme of which has over 400 pages), which is the cause of the apparent discrepancy in variance.

Table 3: Test of Normality (Shapiro-Wilk) of numeric features, showing that a standard t-test would not be appropriate

|  | Reproduced | W | p-value |
|---|---|---|---|
| Number of References | No | 0.920 | $2.583 \times 10^{-5}$ |
|  | Yes | 0.634 | $1.824 \times 10^{-18}$ |
| Normalized Num References | No | 0.953 | 0.002 |
|  | Yes | 0.848 | $1.084 \times 10^{-11}$ |
| Normalized Number of Equations | No | 0.841 | $1.366 \times 10^{-8}$ |
|  | Yes | 0.816 | $5.417 \times 10^{-13}$ |
| Normalized Number of Proofs | No | 0.654 | $1.757 \times 10^{-13}$ |
|  | Yes | 0.671 | $1.454 \times 10^{-17}$ |
| Normalized Total Tables and Figures | No | 0.903 | $3.833 \times 10^{-6}$ |
|  | Yes | 0.722 | $3.608 \times 10^{-16}$ |
| Normalized Number of Tables | No | 0.710 | $2.990 \times 10^{-12}$ |
|  | Yes | 0.885 | $6.970 \times 10^{-10}$ |
| Normalized Number of Graphs/Plots | No | 0.842 | $1.539 \times 10^{-8}$ |
|  | Yes | 0.659 | $7.344 \times 10^{-18}$ |
| Normalized Number of Other Figures | No | 0.632 | $6.193 \times 10^{-14}$ |
|  | Yes | 0.353 | $8.797 \times 10^{-24}$ |
| Normalized Conceptualization Figures | No | 0.572 | $4.755 \times 10^{-15}$ |
|  | Yes | 0.606 | $4.003 \times 10^{-19}$ |
| Pages | No | 0.789 | $3.090 \times 10^{-10}$ |
|  | Yes | 0.697 | $7.302 \times 10^{-17}$ |

Table 4: Test of Equality of Variances (Levene's) for numeric features.

|  | F | df | p |
|---|---|---|---|
| Year | 5.811 | 1 | 0.017 |
| Year Attempted | 0.443 | 1 | 0.506 |
| Pages | 5.299 | 1 | 0.022 |
| Normalized Num References | 2.179 | 1 | 0.141 |
| Normalized Number of Equations | 0.691 | 1 | 0.406 |
| Normalized Number of Proofs | 3.343 | 1 | 0.069 |
| Normalized Number of Tables | 0.260 | 1 | 0.610 |
| Normalized Number of Graphs/Plots | 0.192 | 1 | 0.662 |
| Normalized Number of Other Figures | 0.154 | 1 | 0.695 |
| Normalized Conceptualization Figures | 0.095 | 1 | 0.758 |
| Number of Authors | 0.079 | 1 | 0.779 |
| Normalized Total Tables and Figures | 0.452 | 1 | 0.502 |

## C   Plots of Numeric Features

Figure 1: Histograms of the unnormalized numeric variables considered.

Figure 2: Histograms of the page normalized numeric variables considered.

# D Contingency Tables for Nominal Features

Table 5: $\chi^2$ test for Venue Type ($p = 0.502$) counts and expectations for a paper's Readability towards ability to reproduce its results.

| Reproduced | | Tech Report | Workshop | Conference | Journal | Book |
|---|---|---|---|---|---|---|
| | | | | Type | | |
| No | Count | 1.00 | 1.00 | 58.00 | 31.00 | 2.00 |
| | Expected count | 2.55 | 0.73 | 53.98 | 32.09 | 3.65 |
| Yes | Count | 6.00 | 1.00 | 90.00 | 57.00 | 8.00 |
| | Expected count | 4.45 | 1.27 | 94.02 | 55.91 | 6.35 |

Table 6: $\chi^2$ test for Author's Code being made available ($p = 0.184$) counts and expectations for a paper's Readability towards ability to reproduce its results.

| Reproduced | | No | Yes |
|---|---|---|---|
| | | Author Code Available | |
| No | Count | 49.00 | 44.00 |
| | Expected count | 43.40 | 49.60 |
| Yes | Count | 70.00 | 92.00 |
| | Expected count | 75.60 | 86.40 |

Table 7: $\chi^2$ text for whether an Author Replied to email questions ($p = 6.016 \times 10^{-8}$) counts and expectations for a paper's Readability towards ability to reproduce its results.

| Reproduced | | No | Yes |
|---|---|---|---|
| | | Authors Reply | |
| No | Count | 23.00 | 4.00 |
| | Expected count | 12.96 | 14.04 |
| Yes | Count | 1.00 | 22.00 |
| | Expected count | 11.04 | 11.96 |

Table 8: $\chi^2$ tests for Rigor vs Empirical ($p = 1.545 \times 10^{-9}$) counts and expectations for a paper's Readability towards ability to reproduce its results.

| Reproduced | | Rigor vs Empirical | | |
| | | Empirical | Theory | Balance |
| --- | --- | --- | --- | --- |
| No | Count | 14.00 | 53.00 | 26.00 |
| | Expected count | 29.18 | 30.64 | 33.19 |
| Yes | Count | 66.00 | 31.00 | 65.00 |
| | Expected count | 50.82 | 53.36 | 57.81 |

Table 9: $\chi^2$ tests for a paper having an Appendix $p = 0.330$ counts and expectations for a paper's Readability towards ability to reproduce its results.

| Reproduced | | Has Appendix | |
| | | No | Yes |
| --- | --- | --- | --- |
| No | Count | 52.00 | 41.00 |
| | Expected count | 56.16 | 36.84 |
| Yes | Count | 102.00 | 60.00 |
| | Expected count | 97.84 | 64.16 |

Table 10: $\chi^2$ tests for when a paper "Looks Intimidating" ($p = 0.829$) counts and expectations for a paper's Readability towards ability to reproduce its results.

| Reproduced | | Looks Intimidating | |
| | | No | Yes |
| --- | --- | --- | --- |
| No | Count | 49.00 | 44.00 |
| | Expected count | 50.33 | 42.67 |
| Yes | Count | 89.00 | 73.00 |
| | Expected count | 87.67 | 74.33 |

Table 11: $\chi^2$ test ($p = 9.681 \times 10^{-25}$) counts and expectations for a paper's Readability towards ability to reproduce its results.

| Reproduced | | Paper Readability | | | |
| | | Low | Ok | Good | Excellent |
| --- | --- | --- | --- | --- | --- |
| No | Count | 61.00 | 24.00 | 8.00 | 0.00 |
| | Expected count | 27.35 | 20.79 | 28.45 | 16.41 |
| Yes | Count | 14.00 | 33.00 | 70.00 | 45.00 |
| | Expected count | 47.65 | 36.21 | 49.55 | 28.59 |

Table 12: $\chi^2$ tests for an Algorithm's Difficulty ($p = 2.939 \times 10^{-5}$) counts and expectations for a paper's Readability towards ability to reproduce its results.

| Reproduced | | Algorithm Difficulty | | |
| | | Low | Medium | High |
| --- | --- | --- | --- | --- |
| No | Count | 21.00 | 38.00 | 34.00 |
| | Expected count | 37.56 | 32.09 | 23.34 |
| Yes | Count | 82.00 | 50.00 | 30.00 |
| | Expected count | 65.44 | 55.91 | 40.66 |

Table 13: $\chi^2$ tests for whether a paper has Pseudo-Code ($p = 2.308 \times 10^{-4}$) counts and expectations for a paper's Readability towards ability to reproduce its results.

| Reproduced | | Pseudo Code | | | |
| | | No | Step-Code | Yes | Code-Like |
| --- | --- | --- | --- | --- | --- |
| No | Count | 21.00 | 34.00 | 35.00 | 3.00 |
| | Expected count | 28.81 | 20.79 | 37.20 | 6.20 |
| Yes | Count | 58.00 | 23.00 | 67.00 | 14.00 |
| | Expected count | 50.19 | 36.21 | 64.80 | 10.80 |

Table 14: $\chi^2$ tests for Data being Available ($p = .558$) counts and expectations for a paper's Readability towards ability to reproduce its results.

| Reproduced | | Data Available No | Data Available Yes |
|---|---|---|---|
| No | Count | 17.00 | 75.00 |
| | Expected count | 14.85 | 77.15 |
| Yes | Count | 24.00 | 138.00 |
| | Expected count | 26.15 | 135.85 |

Table 15: $\chi^2$ tests for use of an Exemplar Toy Problem ($p = 0.720$) counts and expectations for a paper's Readability towards ability to reproduce its results.

| Reproduced | | Uses Exemplar Toy Problem No | Uses Exemplar Toy Problem Yes |
|---|---|---|---|
| No | Count | 65.00 | 28.00 |
| | Expected count | 66.74 | 26.26 |
| Yes | Count | 118.00 | 44.00 |
| | Expected count | 116.26 | 45.74 |

Table 16: $\chi^2$ tests for Exact Compute Used being specified ($p = 0.257$) counts and expectations for a paper's Readability towards ability to reproduce its results.

| Reproduced | | Exact Compute Used No | Exact Compute Used Yes |
|---|---|---|---|
| No | Count | 76.00 | 17.00 |
| | Expected count | 71.85 | 21.15 |
| Yes | Count | 121.00 | 41.00 |
| | Expected count | 125.15 | 36.85 |

Table 17: $\chi^2$ tests for Hyperparamters being Specified ($p = 8.450 \times 10^{-6}$) counts and expectations for a paper's Readability towards ability to reproduce its results.

| Reproduced | | Hyperparameters Specified No | Hyperparameters Specified Yes | Hyperparameters Specified Partial |
|---|---|---|---|---|
| No | Count | 34.00 | 54.00 | 5.00 |
| | Expected count | 20.06 | 70.02 | 2.92 |
| Yes | Count | 21.00 | 138.00 | 3.00 |
| | Expected count | 34.94 | 121.98 | 5.08 |

Table 18: $\chi^2$ tests for the level of Compute Needed ($p = 2.788 \times 10^{-5}$) counts and expectations for a paper's Readability towards ability to reproduce its results.

| Reproduced | | Compute Needed Desktop | GPU | Server | Cluster |
|---|---|---|---|---|---|
| No | Count | 78.00 | 5.00 | 0.00 | 10.00 |
| | Expected count | 77.68 | 10.58 | 1.09 | 3.65 |
| Yes | Count | 135.00 | 24.00 | 3.00 | 0.00 |
| | Expected count | 135.32 | 18.42 | 1.91 | 6.35 |

Table 19: $\chi^2$ test for Primary Topic ($p = 7.039 \times 10^{-4}$) counts and expectations for a paper's Readability towards ability to reproduce its results.

| Primary Topic | | Reproduced No | Reproduced Yes |
|---|---|---|---|
| Bayesian | Count | 6.00 | 0.00 |
| | Expected count | 2.19 | 3.81 |
| Class Imbalance | Count | 0.00 | 2.00 |
| | Expected count | 0.73 | 1.27 |
| Classification | Count | 2.00 | 8.00 |
| | Expected count | 3.65 | 6.35 |
| Clustering | Count | 10.00 | 14.00 |
| | Expected count | 8.75 | 15.25 |
| Concept Drift | Count | 0.00 | 4.00 |
| | Expected count | 1.46 | 2.54 |
| Decision Trees | Count | 2.00 | 2.00 |
| | Expected count | 1.46 | 2.54 |
| Deep Learning | Count | 1.00 | 27.00 |
| | Expected count | 10.21 | 17.79 |
| Dimension Reduction | Count | 4.00 | 4.00 |
| | Expected count | 2.92 | 5.08 |
| Embedding | Count | 1.00 | 1.00 |
| | Expected count | 0.73 | 1.27 |
| Ensembling | Count | 7.00 | 13.00 |
| | Expected count | 7.29 | 12.71 |
| Fairness | Count | 4.00 | 0.00 |
| | Expected count | 1.46 | 2.54 |
| Feature Engineering | Count | 3.00 | 5.00 |
| | Expected count | 2.92 | 5.08 |
| Feature Importanace | Count | 0.00 | 1.00 |
| | Expected count | 0.36 | 0.64 |
| Graph Classification | Count | 1.00 | 1.00 |
| | Expected count | 0.73 | 1.27 |
| Kernel/SVMs | Count | 16.00 | 20.00 |
| | Expected count | 13.13 | 22.87 |
| Linear Models | Count | 8.00 | 6.00 |
| | Expected count | 5.11 | 8.89 |
| Meta | Count | 0.00 | 4.00 |
| | Expected count | 1.46 | 2.54 |
| NLP | Count | 1.00 | 3.00 |
| | Expected count | 1.46 | 2.54 |
| Non-Linear Other | Count | 1.00 | 3.00 |
| | Expected count | 1.46 | 2.54 |
| Online Classification | Count | 3.00 | 12.00 |
| | Expected count | 5.47 | 9.53 |
| Optimization | Count | 6.00 | 8.00 |
| | Expected count | 5.11 | 8.89 |
| Other | Count | 2.00 | 2.00 |
| | Expected count | 1.46 | 2.54 |
| Outlier Detection | Count | 3.00 | 1.00 |
| | Expected count | 1.46 | 2.54 |
| Parallel Learning | Count | 3.00 | 2.00 |
| | Expected count | 1.82 | 3.18 |
| Search/Retrieval | Count | 5.00 | 17.00 |
| | Expected count | 8.02 | 13.98 |
| Topic Modeling | Count | 4.00 | 2.00 |
| | Expected count | 2.19 | 3.81 |

## Footnotes

[6]e.g., see the GMP project as an example of a far more robust and similar project `https://gmplib.org/`