[Reviews · NeurIPS 2019]

Reviewer 1



The paper presents the outcome of reproduction efforts of 255 prior studies, analyzing the relation of success of reproduction and approximately 25 features (some quantitative, some more subjective/qualitative) extracted from these papers. The authors present and discuss the features that predict reproducibility significantly, as well as those that are not found significantly effective despite (authors') expectations (e.g., conceptualization figures, or step-by-step examples presented in the paper). The paper is well written, and I found the discussion insightful. The analyses of the data is based on traditional statistical testing. Despite some shortcomings (e.g., biased sample - also acknowledged by the authors), this is a rather large-scale study, and it contributes to the ongoing discussion on replicability/reproducibility in the field. The main weakness of the study is the biased sample. ------- After author response: Thank you for your response. Additional discussion/information suggested will improve the paper, and my "automatic text classification" suggestion was not realistic to incorporate in this paper anyway. After reading other reviews and author response, I (still) think that this paper provides valuable data and may stimulate important/useful discussion in the field. I believe it should be accepted.

Reviewer 2



The working topic on evaluating paper reproducibility is important for an increasing research community. This paper conducts an empirical study on what impacts the reproducibility of research papers. However: 1) The definition to reproducibility is vague. They say that a paper is reproducible if they managed to reproduce its main claims. What does it mean exactly? Come close to reported performance scores? Beat competitors? 2) How is it possible that you reimplement the codes of 255 papers independently without looking authors’ code? This appears as an enormous labor effort. The paper is mostly clear and has good quality. I am not entirely sure about its significance for NIPS as the topic is not directly concerned with machine learning, but rather with a meta-introspection. One might think that a workshop like https://sites.google.com/view/icml-reproducibility-workshop/home could be a suitable venue as well.

Reviewer 3



This is a well-written and carefully researched paper on what factors in a paper are correlated with it being more or less reproducible. Reproducible here means the authors were able to independently author the same algorithm and reproduce the majority of the claims as the original paper. The statistical analysis is careful and complete, and the discussion of the results is nice. It would be interesting to note not only whether the paper was reproducible, but how long it too to reproduce (scaled by "size" of the thing being reproduced. Perhaps something like days to implement divided by lines of code in the implementation. The authors mention that extracting the features per paper took only about 30 minutes, but didn't say how long implementation took, other than mentioning one case that took over 4 years. It would be interesting to see the distribution of how long it took to reimplement. It would be nice if the paper had a bit more discussion about how it was decided whether an implementation successfully reproduced the claims of the paper. I'm assuming you did not require getting exactly the same results down to the hundredth of a percent, but presumably required getting "similar" results? Some more clarity here would be helpful. There's really nothing wrong with this paper. I think it's a good, solid contribution and valuable to the field. The only potential issue is whether it may fit better at a different conference focused more on the science of science, but really I think it fits fine at NeurIPS too. I read the author response and agree with their assesment that it is reasonable to publish this paper at a conference like NeurIPS due to the ML focus of the papers being reproduced.

[Author Response · NeurIPS 2019]

Thank you all for the thoughtful reviews! We will respond in plural form to maintain blindness. We will first address issues brought up by multiple reviewers, followed by individual queries.

**Regarding the definition of reproducibility:** We regarded a paper as reproducible if most (75%+) of the claims in the paper could be reproduced. If a claimed improvement was measured in orders-of-magnitude, being within the same order-of-magnitude was sufficient (e.g., a paper claims 700x faster, but we observer 300x, still qualifies). When compared to other algorithms, we consider a paper reproduced if most (90%+) of the new algorithm's rankings correspond to what was in the paper (e.g., new method was most accurate on 95% of tasks compared to 4 other models, we want to see our reproduction be most accurate on at least 95%*90%=81% of the same tasks, compared to the same models). As a last resort, we considered getting within 10% of the numbers reporting in the paper (or better), or in the case of non-quantitative results (e.g., GAN sample quality), we subjectively compare our results with the paper to make a decision. We will include a version of the above in the revised paper.

**Regarding Venue:** While a "science of science" type venue would be appropriate, we feel NeurIPS is a more appropriate venue. We are more concerned with the nature of reproducibility within our specific field, rather than broader genres such as computer science or science generically. We feel the discussion appropriate given the current high and growing discussion of these issues within the field, and necessary to build communal momentum around larger organized efforts to track and quantify reproduction (as we discussed regarding the ICLR Reproducability Challenge). While we consider our work valuable, issues in study bias will persist until we get a larger pool of implementers under consideration.

**Regarding Author/reproducer details:** We are trying to maintain double-blindness as much as possible in our replies. The camera ready will detail this and reproducer background extensively.

**To R#1:** We would have also liked to do a textual analysis of papers, and hope to do so in the future. However, first attempts produced significant issues with respect to parsing reliability (some papers are photo-copies, some are OCRed, some are poorly formated PDFs, some are nicely formtated PDFs, etc) that would confound results and increase analysis difficulty, not to mention take considerable time to get working reliably. This also prevented us from automated the equation counting.

Re line 212: you make an excellent point, and we will add caution to the final manuscript regarding that statement.

"*Furthermore, the reader would also benefit from a rough classification of reasons for non-reproducible papers*": We did not record this in detail for each paper, but agree would be valuable! We will add a discussion of the general types of issues we encountered in the final version (though unfortunately non-quantified). Subjectively, the below list would be the primary issues that gave us reproduction problems, which we will elaborate further in camera ready. None of the below issues were mutually exclusive.

1. Unclear notation or language. A component of the algorithm is explained, but not in a way easily understood by the reproducers, or was ambigiously specified.
2. Missing algorithm step or details. A step was completely left out of description.
3. Results left as an exercise to the reader: many papers would specify loss functions or other equations for which the gradient needed to be taken, but then not detail the resulting gradients. Depending on the functions and math involved re-deriving was non-trivial.
4. Missing hyper-parameters, or similar nuance details. We appear to have an implementation accurate to what was described, but some minor detail was not specified and makes a big difference in results.

We understand knowing the number of reproducers / backgrounds is intrinsically valuable for our paper, but also violates double blind review. Reproduction attempts/effort was approximately uniformly distributed between reproducers.

**To R#2:** The effort was indeed significant, and only possible because we used software early on that recorded much relevant information. Between efforts done as part of education, job, and for fun, back-of-envelop guesstimation puts our effort at $\approx$10,710 hours per author.

**To R# 3:** On computing "*how long it too to reproduce*", this is a very good idea, thank you! We think we could do this for *most* of our papers that were reproduced, as most where made open source and we can compare our start date to the commit date to get an approximation. We can't do it for all though as some remain closed source, and we would be unable to compare to unreplicated papers since we did not track that information.

We can't get this done by the end of rebuttal time, but will consider it for camera ready - we aren't sure how long this would take us (given day jobs as well!). We also want to give it the same thought and consideration we gave the factors already in the paper, and not rush analysis without considering confounding factors.

[Meta-Review · NeurIPS 2019]

This paper represents a massive amount of work that will have significant impact on research practices in ML. Reviewers are strongly positive, with the only concern being that this is about ML practice and not ML itself. I do not share this concern.